# Of Fire and Smoke Plumes, Polarimetric Radar Characteristics

**Dusan Zrnic [1,2,*], Pengfei Zhang [3] 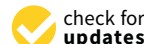, Valery Melnikov [3] and Djordje Mirkovic [3]**

[1]  National Severe Storms Laboratory, National Oceanic and Atmospheric Administration (NOAA), Norman, OK 73072, USA

[2]  School of Meteorology and El. Engineering, University of Oklahoma, Norman, OK 73019, USA

[3]  Cooperative Institute for Mesoscale Meteorological Studies, National Weather Center, Norman, OK 73072, USA; pengfei.zhang@noaa.gov (P.Z.); valery.melnikov@noaa.gov (V.M.); djordje.mirkovic@noaa.gov (D.M.)

*  Correspondence: dusan.zrnic@noaa.gov

**Abstract:** Weather surveillance radars routinely detect smoke of various origin. Of particular significance to the meteorological community are wildfires in forests and/or prairies. For example, one responsibility of the National Weather Service in the USA is to forecast fire outlooks as well as to monitor wildfire evolution. Polarimetric variables have enabled relatively easy recognitions of smoke plumes in data fields of weather radars. Presented here are the fields of these variables from smoke plumes caused by grass fire, brush fire, and forest fire. Histograms of polarimetric data from plumes contrast these cases. Most of the data are from the polarimetric Weather Surveillance Radar 1988 Doppler (WSR-88D aka NEXRAD, 10 cm wavelength); hence, the wavelength does not influence these comparisons. Nevertheless, in one case, simultaneous observations of a plume by the operational Terminal Doppler Weather Radar (TDWR, 5 cm wavelength) and a WSR-88D is used to infer backscattering characteristic and, hence, sizes of dominant contributors to the returns. To interpret these measurements, Computational Electromagnetics (CEM) tools are applied. For one wildfire from Oklahoma, radar and satellite (GOES-16, Geostationary Operational Environmental Satellite) images are analyzed. The case demonstrates a potential to forecast fire intensification caused by a very rapid cold front. Finally, we suggest a possible way to extract the smoke plume return from the class of nonmeteorological scatterers.

**Keywords:** weather radar; polarimetry; smoke plumes; wild fires; polarimetric characteristics

## 1. Introduction

From 1992 to present, the United States Forest Service has a record of geo-referenced wildfires in the USA. From their statistics [1], Hoover and Hanson write that since the year 2000 "an average of 72,400 wildfires burned an average of 7.0 million acres per year." This is about the land size of Maryland. The minima of burned land are about 4.0 million acres, or 6250 square miles, and is close to the land size of Hawaii. The maxima are 10 million acres, which is about twice the land size of Massachusetts. Besides surface damage, the wild fires caused between 12 and 19 firefighters fatalities per year from 2015 to 2018. To reduce the burned acreage and fatalities, early detection of wild fires is imperative. In areas void of population the principal wildfire detecting instruments are satellites and weather radars.

According to the US National Park Service [2] about "85% of wildland fires in the United States are caused by humans". These include fires started as campfires and left unattended, burning of debris, use of equipment and malfunctions, discarded cigarettes, and arson. Nevertheless, the underlining conditions for recent increase in the occurrence of wild fires is the raising temperature of the atmosphere.

The rise is expected to continue and research projects increased "trend towards more dangerous near-surface fire weather conditions in Australia and pyroconvection risk factors" [3].

Weather surveillance radars routinely observe smoke plumes of various origin. Their high sensitivity enables detecting and tracking plumes [4], and helps management of wildfires [5]. Detection of wildfires by weather radar is on average 5 min after ignition compared to the 15 min delay achieved with human observers [6]. Researchers used operational radars to infer injection heights of smoke aerosols in Southern Georgia, USA, by associating these to the heights of detectable ash particle [7]. The study finds mean height of $3 \pm 1$ km occurs in late afternoon when the fire and convective mixing are strongest. Nonetheless, in that study there was no clear indication of pyroconvection. Studies of pyroconvection include observation of ash, ice, and lightning [8] in which the authors are able to separate ash from ice particles using the polarimetric variables. Radar observations of pyroconvection combined with lightning mapping array indicates that lightning occurred whenever the smoke plume grew to 10 km [9] of mean sea level. These scientists capitalized on the polarimetric data (CSU CHILL, Colorado State University and Chicago Illinois radar) to identify the smoke part of plumes by noting low reflectivities (10–25 dBZ), increased $Z_{DR}$ (1–5 dB), and small correlation coefficient between returns at orthogonal polarizations, $\rho_{hv}$ of about 0.6.

Reference [10] contains the polarimetric characteristics of a grass fire, whereas in [11], scientists have used mobile 3-cm wavelength radar to investigate pyroconvection and wildfire meteorology. That was the first coordinated field project to study the "fire-atmosphere dynamics". Its operation relied on forecasts similar to storm chase. The reference contains detailed analysis of the fire dynamics and evolution. The relevant information for our study is the probability density function of the differential reflectivity and the correlation coefficient observed in the plume and rain [11], as we can compare these to our histograms.

A comprehensive review of wildfire observations with radars is in [12]. The authors introduce the term pyrometeor for the ash particles causing the returns and advocate "radar research to establish the cross section and dielectric factor $K_m$ for scatterers of pyrogenic origin". These characteristics are extremely hard to determine from in situ observations. Nevertheless, measurements in a laboratory setting are feasible and several studies document these [13–15]. Most consider the frequency band 8–12 GHz, but [15] considers 10 GHz and 38 GHz, whereas [16] covers the 8–12 GHz and 26.5 to 40 GHz range. We are not aware of such studies for the 10 cm wavelengths, although laboratory measurements of materials that might be similar to ash exist [17].

Weather radars also detect smoke from urban areas. A good example is the industrial fire in Montreal observed with a 10 cm wavelength weather radar [18]. The authors document the history of the plume and compare simultaneous observations with a vertically pointing 3 cm wavelength radar and a 33 cm wavelength wind profiler. The reflectivities $Z$, measured with the 33 cm wavelength radar, reach 40 dBZ while those measured at the 3 cm wavelength are about 20 dB lower. One explanation is that the particle sizes, approximately 1 cm, caused Mie scattering at the 3 cm wavelength while at the 33 cm wavelength the scattering was in the Rayleigh regime, characterized with a significantly larger cross section and, consequently, stronger reflectivity $Z$. In addition, the authors [18] hypothesize that refractive index irregularities also contributed to the difference. However, others [19] suggest that coherent scattering from the particles in smoke may be significant and would explain the correlation between the reflectivities at the two wavelengths. In [20], the authors write about dual polarization characteristics, at 5 cm wavelength, of an apartment fire. They found mean reflectivity of 9 dBZ within the plume and maximum values of 20 dBZ. Their mean differential reflectivity, $Z_{DR}$ is 1.7 dB, similar to values in rain, but the low correlation coefficient (less than 0.5) clearly indicates nonmeteorological scatterers.

In this paper, we document polarimetric radar observations of smoke caused by wildfires fueled by different vegetation. Of particular significance to the meteorological community are forests and/or prairies. For example, the US National Weather Service (NWS) Storm Prediction Center issues daily fire weather outlooks. Forecasters at local offices have access to display of weather radar data in which

they can identify and track smoke plumes. They issue fire weather watches and red flag warnings. Related is the potential for mudslides and debris flow on steep terrain made barren by wildfires. Predicting these events is at the core mission of the NWS and researchers suggest development of a comprehensive system to combine radar, satellite, and models for fire weather forecasting [21].

We present polarimetric characteristics of smoke from two grass fires in Oklahoma, forest fire in New Mexico, and brush fire near Los Angeles, California. We use data from Weather Surveillance Radar 1988 Doppler (WSR-88D), which have a wavelength of 10 cm. To estimate dominant scatterers' size, we compare the reflectivities from the Twin Lakes, Oklahoma WSR-88D with those observed with the Terminal Doppler Weather Radar (TDWR), which surveys the Oklahoma City airport. That radar's wavelength is 5 cm.

GOES-16 (Geostationary Operational Environmental Satellite) data at 1 min intervals are available for one Oklahoma grassfire. A strong front, observed with a WSR-88D, blew over the fire. This gave us opportunity to compare the observations by these two operational systems.

We use the polarimetric properties of wildfires to construct a rudimentary classification method for identifying wildfires. This we do by using an existing classifier of radar returns into meteorological and nonmeteorological origin [22] and adding to it the wildfire class.

## 2. Examples of Observed Smoke Plumes

This section describes observations of smoke plumes from four wildfires.

### 2.1. Grassfire in Oklahoma, February 12, 2017

In 2017, Oklahoma experienced a dry spell [23], which contributed to several wildfires. One started late in the morning on February 12, 2017, 10 to 20 km southwest from the Oklahoma City Operational WSR-88D (Figure 1). The radar has dual polarization, wavelength $\lambda = 10$ cm, beamwidth $1°$, sample spacing 250 m, and range resolution 235 m.

Figure 1 displays the fields of radar variables obtained from this fire. Background patterns indicate convective rolls, which form when low-level air in the planetary boundary layer is unstable but capped by a stable layer. The polarimetric variables are typical of insects. Notable in the plume are positive differential reflectivities ($Z_{DR}$) of about 2 dB, low correlation coefficients ($\rho_{hv}$) of about 0.6, large differential phases ($\Phi_{DP}$), smooth Doppler velocity field of about 20 m s$^{-1}$, and consistent spectrum widths ($\sigma_v$) of about 2 m s$^{-1}$. The system differential phase on the WSR-88D, $\Phi_{DPsys}$ is $60°$ hence the backscatter differential phase ($\delta = \Phi_{DP} - 60°$) spans a very large range (Figure 2) exceeding that of birds, which can be $0°$ to $120°$ [24]. The Doppler velocity field shows northeast wind at about 10 m s$^{-1}$ and confirms that the smoke particles are very good wind tracers. The spectrum widths in the plume are on the average 1 m s$^{-1}$, and in the environment, these are 2 m s$^{-1}$ (Figure 2). The difference we attribute to the geometry: the plume is aligned with the roll and the beam is almost parallel to the roll's axis. Therefore, the rotation components of rolls are nearly perpendicular to the beam axis and contribute minimally to the spread of Doppler velocities within the resolution volume. The histogram of $\sigma_v$ from the environment is a bit wider because data from all azimuths are included, increasing its mean value and width.

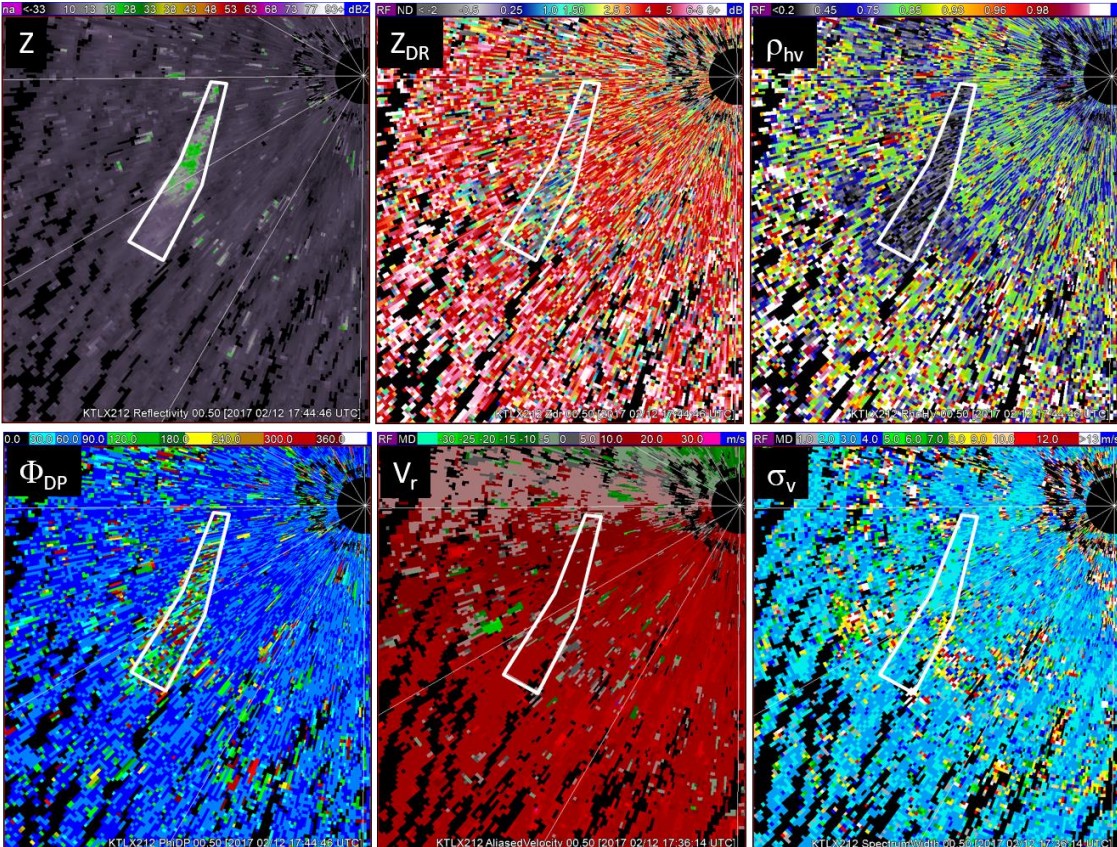

**Figure 1.** Fields of reflectivity, differential reflectivity, correlation coefficient, differential phase, Doppler velocity, and spectrum width. The radar is the Weather Surveillance Radar 1988 Doppler (WSR-88D) (Twin Lakes, Oklahoma City, OK, code designation KTLX), elevation angle is 0.5°, and the date is February 12, 2017 time 17:45:59 UTC. The color bars indicate dBZ units for reflectivity $Z$, dB for differential reflectivity $Z_{DR}$, degrees for differential phase $\Phi_{DP}$, and m s$^{-1}$ for Doppler velocity $v_r$ and spectrum width $\sigma_v$. The polygons contain data from the plume.

Histogram of $Z_{DR}$ from the plume (Figure 2) depict values from −4 dB to over 8 dB, which is the maximum recordable on the WSR-88Ds. We see a large spread of backscatter differential phase with positive values prevailing. The standard deviation of $\Phi_{DP}$ estimates is about 2.5° and the formula for computing it is in [25], so the values at high $\Phi_{DP}$ are unlikely due to uncertainty of estimates. The probable cause is couplings of the H and V components via canted smoke particles upon backscattering [26]. The simultaneous mode (SHV) of polarimetric measurements is prone to coupling, and the "inferred" (wrong) differential phase depends on the differential phase upon transmission, on orientation of the scatterer, on relative reflections at the two polarization, and on the backscatter differential phase δ [27]. These factors create a wide spread of δ. The histograms of the same variables within and outside the plume (Figure 2) overlap. The best separation between values from smoke and environment is in the histograms of reflectivity and correlation coefficient, but some separation is evident in the other two polarimetric variables, as well. Based on the histograms it is possible to construct fuzzy logic membership functions and/or priory probabilities for Bayesian classifiers.

The plume extent in height is about 1.1 km, which is the top of the boundary layer as can be best seen in the fields of $\rho_{hv}$ (Figure 3).

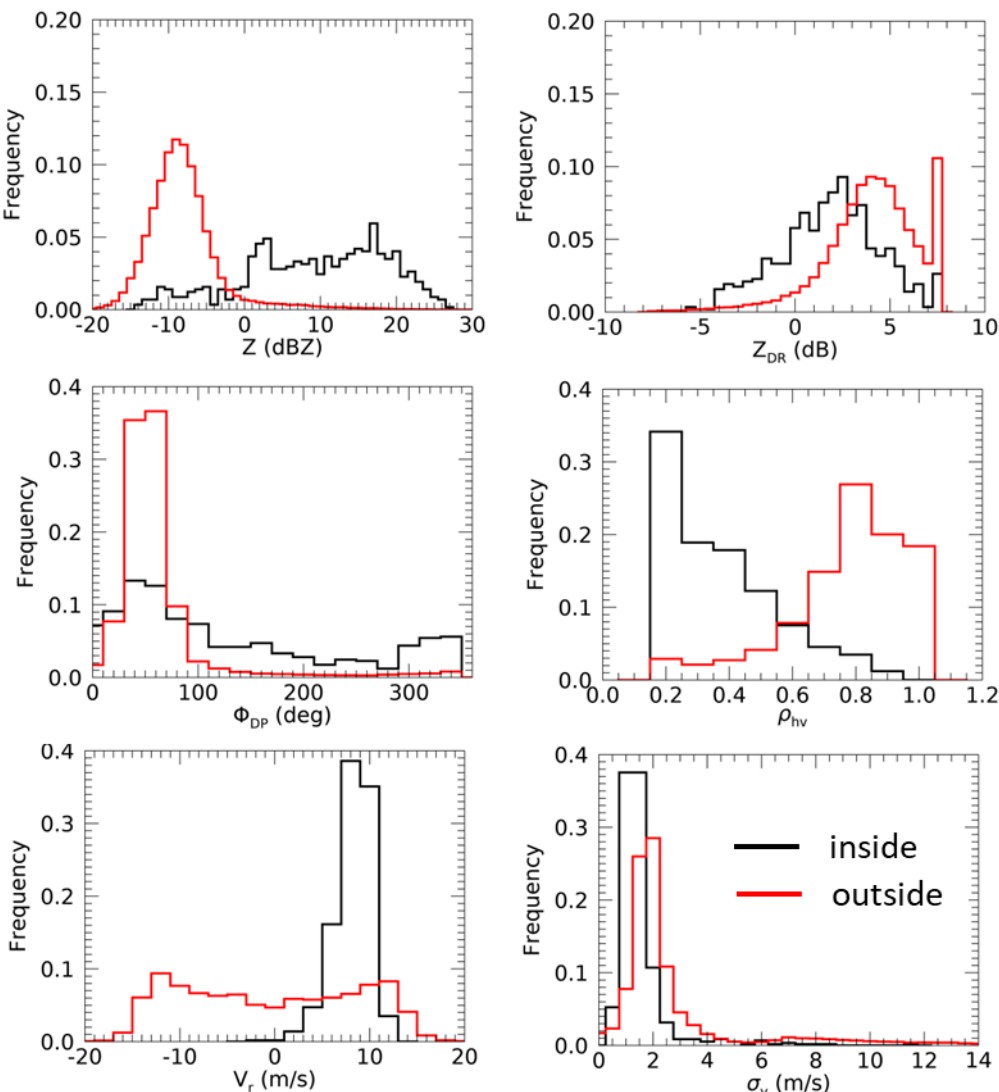

**Figure 2.** Histograms of the polarimetric variables, from the smoke plume (black) and from the area outside of the plume (red). Data are from the scan in Figure 1.

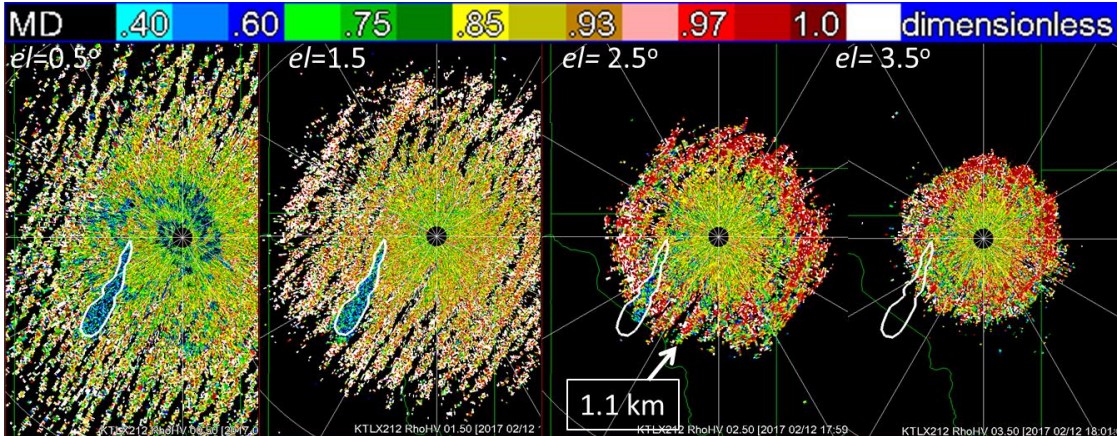

**Figure 3.** Fields of correlation coefficient $\rho_{hv}$ at consecutive scans in elevations of the WSR-88D. The color bar indicates value categories. Date is February 12, 2017.

### 2.1.1. Observation with the Terminal Doppler Weather Radar

The fire was also observed with the 5 cm wavelength (Figure 4), TDWR, which monitors weather hazards over the Oklahoma City Will Rogers Airport. The radar has linear horizontal polarization, beamwidth 0.5°, and sample spacing and range resolution of 150 m. Obvious in these figures are larger reflectivity estimates at the 10 cm wavelength (WSR-88D) compared to the ones at the 5 cm wavelength (TDWR). The histograms of the reflectivities (Figure 5a) quantify this difference and the reason is in the type of scattering, which we discuss shortly. This is in contrast to the regions outside of the polygons from which the histograms are similar (Figure 5 b). The offset of about few dB might be due to Bragg scattering by refractivity irregularities. The top of the boundary layer (BL) was at 1.1 km and within it, the relative humidity was 66%. Above the BL the relative humidity decreased to 20%; mixing of this large gradient could create significant Bragg scattering. The reflectivity $Z$ is proportional to the structure parameter $C_n^2$ of the refractive index fluctuations [28]:

$$\log(C_n^2) = -11.5 + 0.1Z, \tag{1}$$

where $Z$ is in dBZ. These fluctuations are very often present in the boundary layer and can produce reflectivities up to $-3$ dBZ [29] at the 10 cm wavelength. At the 5 cm wavelength, the sizes of the potentially contributing eddies is 2.5 cm, and these are more likely to be in the dissipative range of turbulence than the 5 cm sizes that contribute coherently to the reflectivity at the 10 cm wavelength. Note that most of the $Z$s in the histogram (Figure 5b) from WSR-88D are smaller than $-3$ dBZ and overlap the Bragg scattering values in [29].

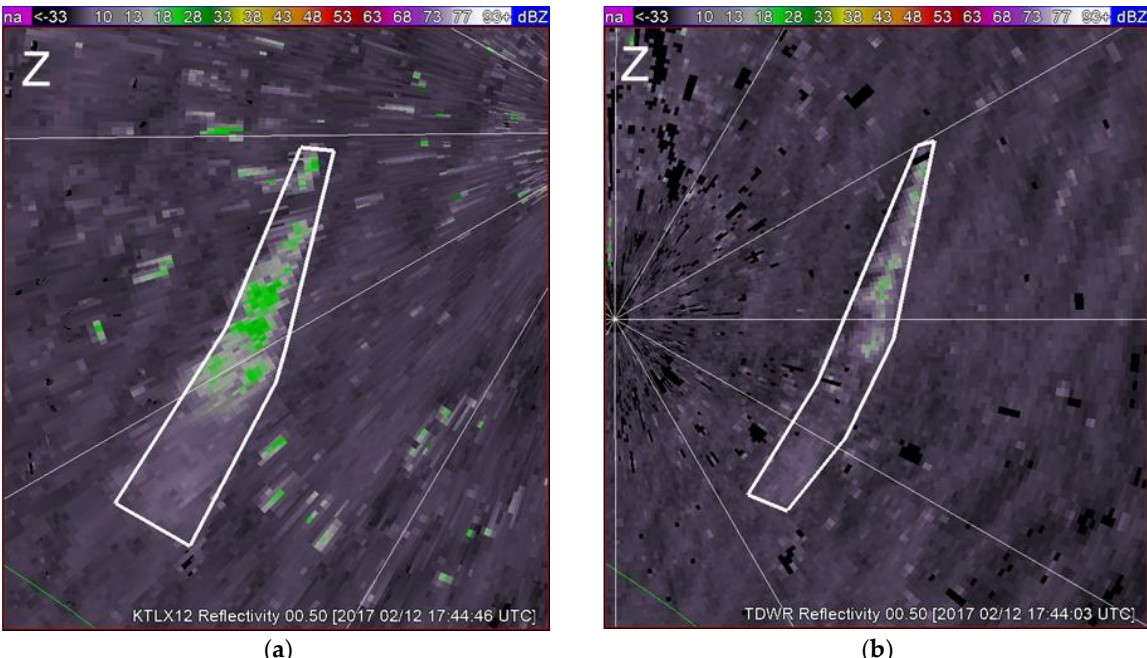

(**a**)             (**b**)

**Figure 4.** Observed reflectivity fields: (**a**), with the Oklahoma City WSR-88D ($\lambda$ = 10 cm, code name KTLX), and (**b**), Terminal Doppler Weather Radar (TDWR) ($\lambda$ = 5.35 cm, code name TOKC) located in Norman for surveillance of the Will Rogers Airport. The color bar indicates reflectivity in dBZ. Date is February 12, 2017 and the polygons encompass the plumes.

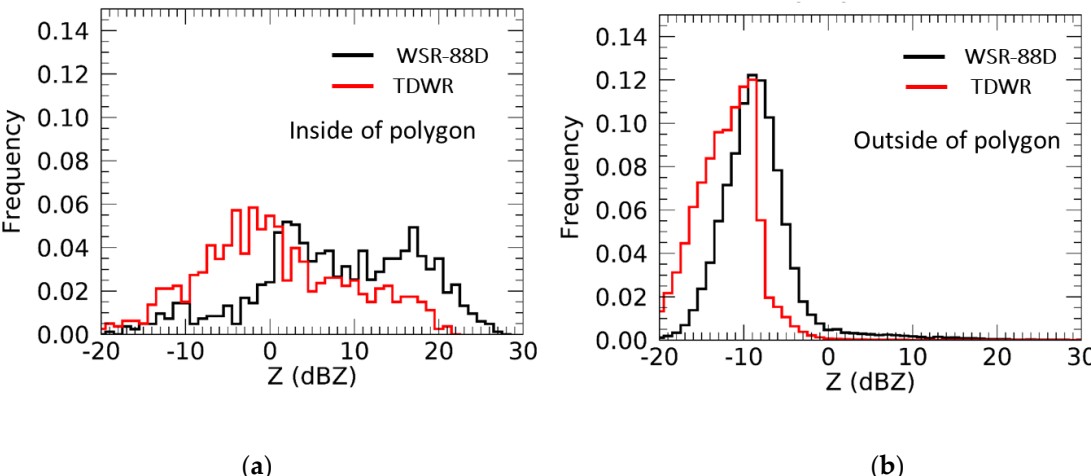

**Figure 5.** Histograms of reflectivity (**a**) within the enclosed areas in Figure 4; black is from the WSR-88D and red is from the TDWR. (**b**) Same as in (**a**) except the histograms are from the areas out of the enclosure. The tail of the distribution from the WSR-88D extends to 30 dBZ, whereas the one from the TDWR extends to 20 dBZ.

As an aside, there are some speckles of Z close to 30 dBZ in the data from WSR-88D (Figure 4a) and none in the data from TDWR (Figure 4b). We speculate that the sporadic spackles are from scatterers that are in the Mie regime at the shorter wavelength but still in the Rayleigh regime at the 10 cm wavelength. They could be birds. Their number is very small and appears in the tail of the distribution (Figure 5b).

We hypothesize that the difference in reflectivities (Figure 5a) off the plume is due to ash debris that is in the Mie regime of scattering at the 5 cm wavelength but still in the Rayleigh at the 10 cm wavelength. During burning of vegetation, the oils burn fist and the water evaporates leaving carbon and minerals. Some carbon may burn into carbon dioxide or monoxide. Often the carbon burning is incomplete, leaving solid residue. In such cases, the biomass (leaves, grass) retains the original shape.

2.1.2. Model of Ash Particles

To test quantitatively the hypothesis that the difference in reflectivities is due to the backscattering regime, we developed a couple of simple backscattering models. Two modes of particles motion exist in plumes. One is the acceding mode within the pyro updraft, in which particles experience strong sheer and turbulence hence exhibit very chaotic motion [15]. Ash particles away from the pyro updraft are in the descending mode and typically exhibit free fall patterns. The prevailing one is fluttering (or swaying) in which ash particle sways back and forth like a pendulum while continuously changing the direction of the sway [15]. Some particles spins about the vertical axis while falling, and some tumble, evolving eventually into fluttering. We have experimentally verified these modes by dropping pieces of dry leaves and observing their motion.

We chose three body types to model the ash particles. One is a flat pentagonal plate (inset in Figure 6a), which models burned pieces of leaves. Similar particles have been observed on windshields of cars. The other two are cylinders, of which, one is hollow to mimic grass, and the other is full to represent burned branch pieces. We chose the pentagon plate with two equal orthogonal dimensions; this accentuates the effect of size (area of plate) and reduces the effects of shape. The thickness is set to 0.15 mm, which is at the low end in [15].

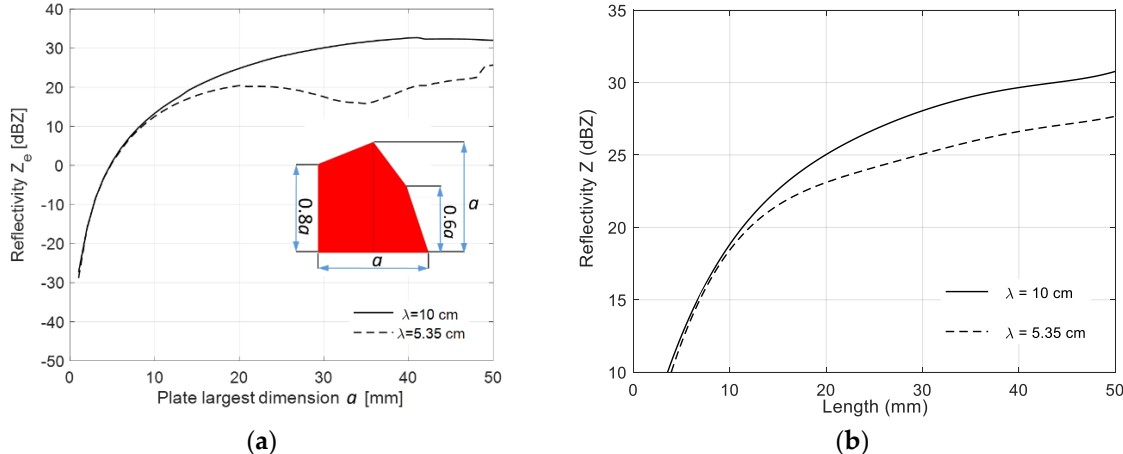

**Figure 6.** (**a**) Reflectivity of a pentagonal plate (inset) representing ashes from vegetation with the thickness of 0.1 mm and permittivity 7 +j2. The concentration is $10^{-1.5}$ m$^{-3}$, and the dimension $a = 1.02D$, where $D$ is a diameter of a circle with the same area as the plate. (**b**) Same as in a) but the model is a hollow cylinder and the concentration is $10^{-1.72}$ m$^{-3}$.

A wide range of dielectric constant (permittivity) may be possible for ash particles and many come from laboratory measurements. The reference [14] lists the largest possible values for five leaf types in Australia. The real parts range from about 4.85 to 17. Moreover, [15] shows curves of permittivity dependence on the volume fraction up to the value of about 0.4. The real part of the effective permittivity is between 2 to 4. Another study [17] presents measurements in the 1 to 10 GHz band, of carbon black at the volume fraction of about 0.1. The value is about $7 + j\,2$, which is what we used to generate Figure 6a.

We computed the reflectivity using the WIPL-D software [30,31]. We specified random orientation of the plate in terms of its yaw (360°), pitch (±30°) and 60° of roll when the roll axis is in the horizontal plane. The model of variables at the 10-cm wavelength accounts for the SHV (simultaneous transmission and reception of the H and V components) polarimetric mode, which is standard on all WSR-88Ds. Therefore, the reflectivity is proportional to $|s_{hh + shv}|^2$, where the second index in the backscattering matrix coefficient stands for the incident polarization (h) and the first indicate the backscattered polarization. The model of reflectivity at the 5-cm wavelength computes only the copolar reflection, which is proportional to $|s_{hh}|^2$ because the TDWR transmits linear horizontally polarized waves. For illustrative purpose, we compared the $Z_e$ with the values in the histogram and applied a concentration that matches the reflectivity (10 cm wavelength at size 20 mm) of 25 dBZ. To do so required subtraction of 15 dBZ from the curves valid at concentrations of 1 m$^{-3}$. This means that the plates' concentration equals $10^{-1.5}$ m$^{-3}$.

The span of plates' reflectivities for which $Z_e(\lambda = 10$ cm$) > Z_e(\lambda = 5$ cm$)$ is from sizes 10 to over 50 mm (Figure 6). According to the histogram (Figure 3a) the difference between these two reflectivities is larger than 5 dB and the graph in Figure 6a shows that such difference can occur at sizes larger than 20 mm.

The difference of reflectivity factors Z(S-band)-Z(C-band) at the size of 20 mm and for the relative dielectric constant $\varepsilon_r$ between 2 and 17 changes from 4.2 to 4.5 dB, which is insignificant. Hence, cannot be used to estimate the true $\varepsilon_r$. Nonetheless, this lack of sensitivity gives more credence to the sizes that can be deduced from the model.

The hollow cylinder model has a diameter of 5 mm and thickness 0.15 mm. We assume the same dynamics as for the plate. The model (Figure 6b) differs significantly from the plate. The concentration that matches the reflectivity at the length of 20 mm with the observation (25 dBZ) is $10^{-1.72}$ m$^{-3}$, which is slightly smaller than the one for the plate. The difference $Z_e(\lambda = 10$ cm$) - Z_e(\lambda = 5$ cm$)$ is at most 3 dB. This is too small to match the observation (Figure 3a). We also modeled small branches as full

cylinders. In that model, the difference between the $Z_e$s at 10 and 5 cm wavelengths was smaller than the one for the hollow cylinder; hence, we do not display it.

The distance from the WSR-88D to the plume's centroid is 15 km and from the TDWR it is 10 km. At these distances and using information in Figures 5a and 6a, we illustrate a possible number of scatterers in the WSR-88D and TDWR resolution volumes. The beamwidths of the WSR-88D and TDWR are 0.95° and 0.5° and the widths of the range weighting functions are 235 m and 150 m. These specify the resolution volume sizes (i.e., volume within which the radar weighting function is equal or larger than 1/4, i.e., −6 dB of its maximum [28]). The corresponding volume sizes are $11.42 \times 10^6$ m$^3$ for the WSR-88D and $0.9 \cdot 10^6$ m$^3$ for the TDWR. Assume the scatterer (plate) largest dimension is 20 mm, so that $Z_e$ = 25 dBZ for the WSR-88D and about 20 dBZ for the TDWR (Figure 6a). If so, there would be about 361,000 scatterers in the resolution volume of the WSR-88D, and 28,300 scatterers in the TDWR's resolution volume. In actuality, the scatterers have a distribution of sizes and may not fill the resolution volumes. Although this challenges quantitative interpretation, the basic conclusion that Mie scattering causes the difference in $Z_e$s stands.

The prairie fires in the southwest are fueled by grasses, forbs (like wild sunflower, milkweed) and possibly red cedar. Having no direct evidence of ash type from the prairies' vegetation we speculate that the "plate" like particles could be from forbs' leaves. The ash origin is a mixture of prairie plants, the dominant contributors to $Z_e$ are the biggest particles, and these could cause the observed difference in the reflectivities (Figure 6a).

## 2.2. Prairie fire Oklahoma, April 18, 2018

We present weather radar observations of a wildfire (Figure 7) that occurred on April 18, 2018 in western Oklahoma. The vegetation consisted of grass and red cedars. Three consecutive fields of reflectivity and matching images from the GOES-16 satellite depict the fire progression. The available GOES images are at 1 min intervals but the radar scans are 10 min apart. In this particular case, the rapid update of satellite data has a clear operational advantage.

Noteworthy are the following features in Figure 7. The satellite image (top) depicts well the fire whereas the Z field does not. The cloud arc on the satellite images and similar arc in the Z fields are indicators of the strong westerly front, which is advancing at 70 km h$^{-1}$. The fire spread and intensified rapidly just as the front blew over it (Figure 7 middle panels). Satellite images two min apart illustrate better this evolution (Figure 8). In four minutes, the fire area (Figure 8, 3:40 to 3:44) grew by more than two times. The smoke became discernible in the Z field at 3:52 and its area is much smaller than the fire area and offset to the southeast (Figure 7 middle panel), undoubtedly due to the strong wind. Ten minutes later (time 4:02 UTC) the fire image exhibits almost no change, whereas the plume has expended by more than ten times.

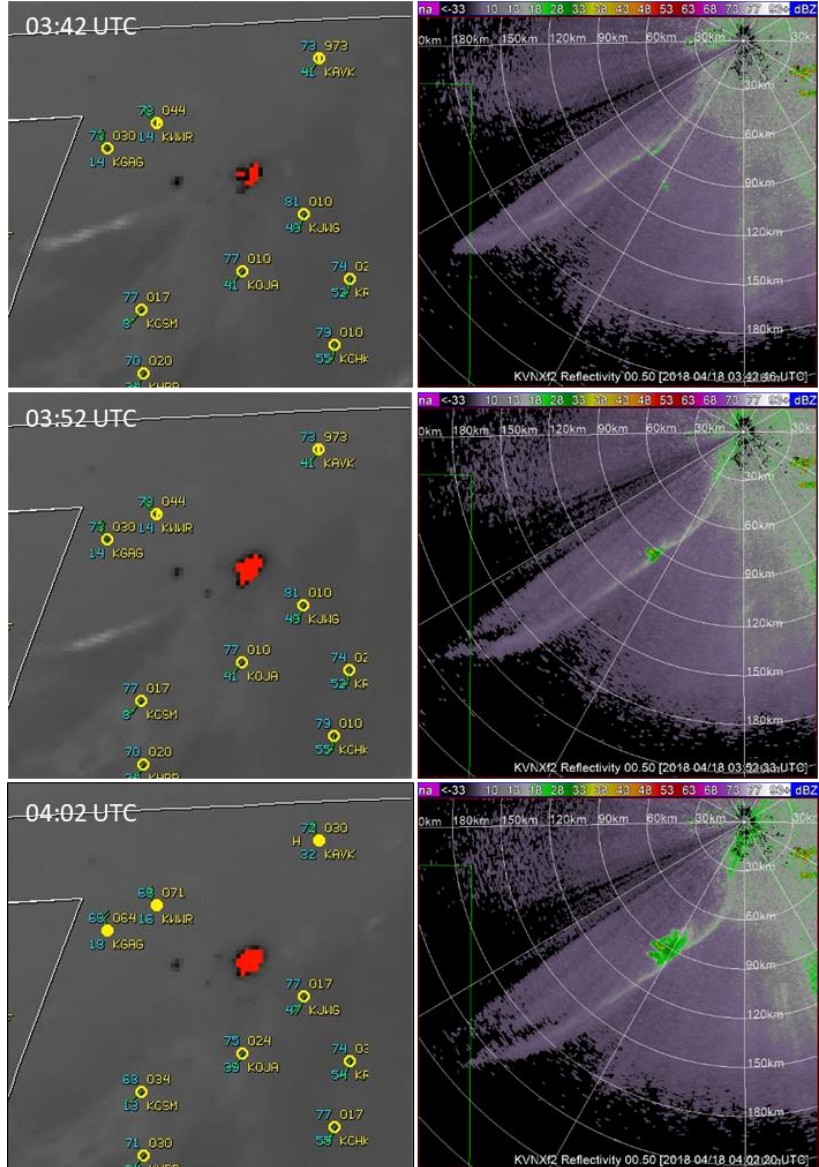

**Figure 7.** (**Left panels**) Shortwave infrared images from the GOES-16 satellite of the April 18, 2017 fire in Oklahoma (red patches depict fire locations). (**Right panels**) Reflectivity fields of the fire taken within 1 min of the satellite images. The color bars indicate Z values in dBZ, and the radar scan is at 0.5° elevation.

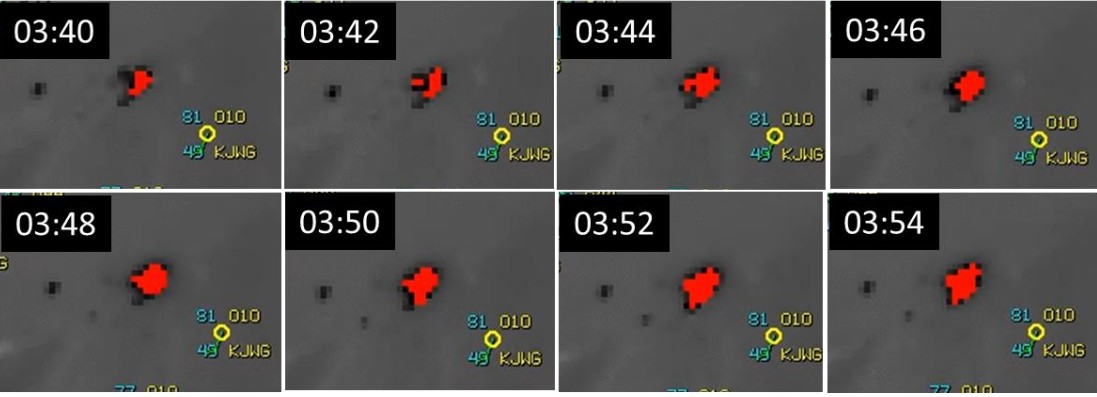

**Figure 8.** Sequence of satellite images from 3:40 to 3:54 UTC. Oklahoma wildfire, April 18, 2017.

The fields of $\rho_{hv}$ at the same times as in Figure 7 are in Figure 9. Just after the frontal passage, the $\rho_{hv}$ decreases (< 0.3).

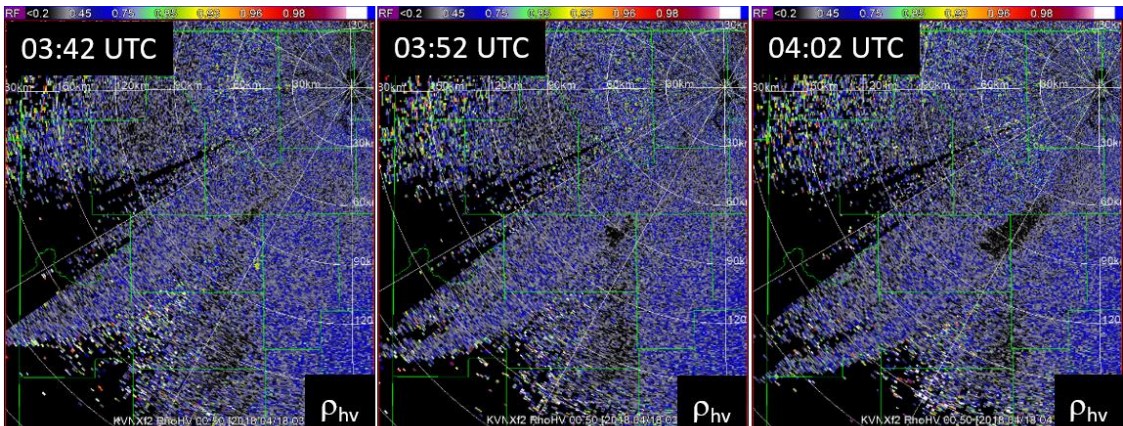

**Figure 9.** Fields of the correlation coefficient at the same times as in Figure 7. The elevation angle is 0.5°.

Vertical cross section (RHI) of $Z$ (Figure 10) exhibits clear separation of the ascending (pyro updraft) region ($Z$ > 33 dBZ) from the advected plume where ash slowly descends. However, in the vertical cross section of the $\rho_{hv}$ field the two regions are indistinguishable. This is unlike the findings in a Florida wildfire [32] where the $\rho_{hv}$ near the updraft is as low as 0.2 and increases downstream.

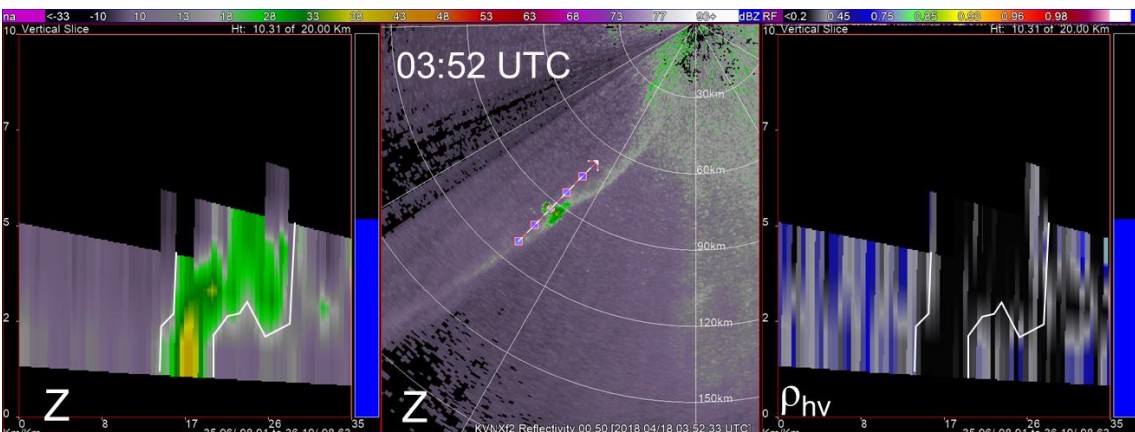

**Figure 10.** (**Left**) Vertical cross-section (RHI) of the $Z$ field at 03:52 UTC. The line in the conical scan (middle panel, 0.5° elevation) indicates the location of the RHI plot. (**Right**) Same as in the left panel but the vertical cross-section of $\rho_{hv}$. One color bar extends over the two images of $Z$ and indicates categories in dBZ. The color bar over the $\rho_{hv}$ field indicates categories, and heights are above ground level in km.

The histograms of the polarimetric variables (Figure 11) from this plume are similar to the ones from February 12, 2017. Although the distance to the plume is about 95 km, its reflectivity reaches 40 dBZ indicating that beam broadening effects if any are secondary. The primary contributors are large ash particles in sufficient numbers filling a good portion of the beam. The differential reflectivity within the plume has almost the same spread as in the February 12 case (Figure 2) and the medians are very close (1.5 to 2.5 dB). The $\rho_{hv}$ histogram is slightly more skewed towards 0.2 than the histogram in Figure 2. The differential phase is centered on the system phase (~60°) and has a large spread.

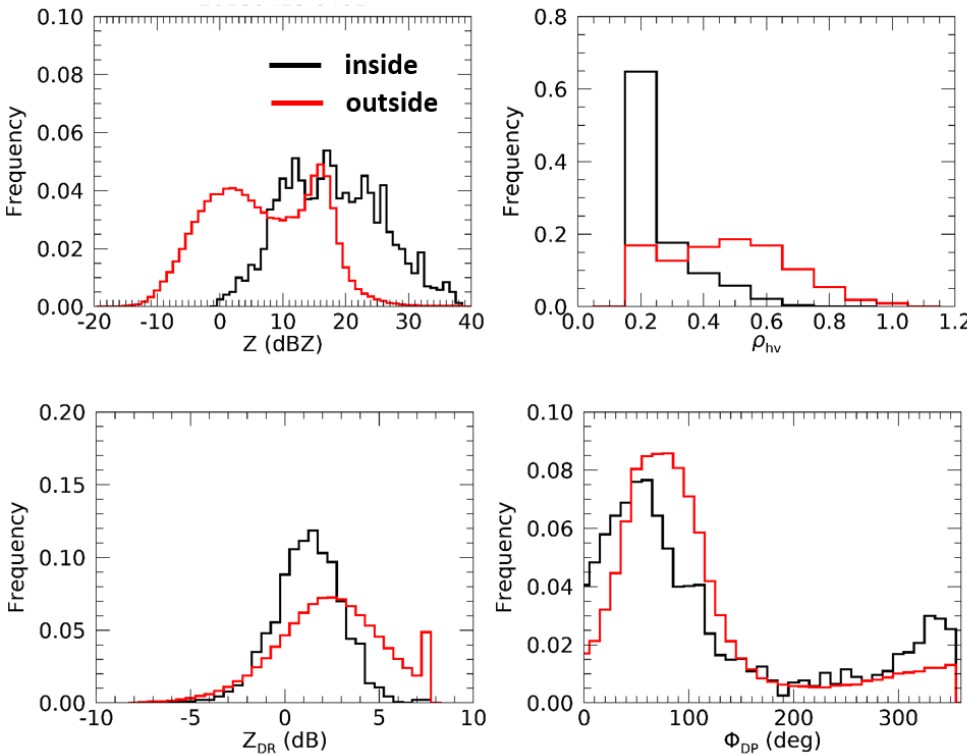

**Figure 11.** Histograms of the polarimetric variables taken from 0.5° elevation scan at 04:02 UTC.

## 2.3. Little Bear Wildfire in New Mexico

On June 4, 2012, lightning ignited a fire in the Little Bear area of the Lincoln National Forest northwest from Ruidoso, New Mexico. By June 8, preliminary defense line was completed around the fire perimeter, but on June 8, strong winds blew fire embers beyond the perimeter. The fire burned more than 44,000 acres, 242 houses, and 12 structures. The photograph in Figure 12 (left) depicts the fire on June 8 and the burnout terrain is in Figure 12 (right). The steep parts of the burnout terrain are prone to mudslides.

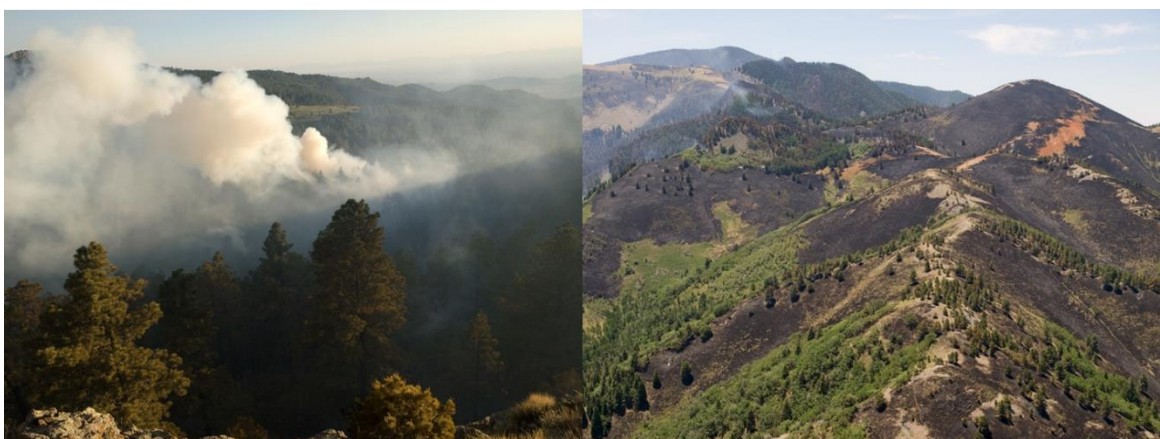

**Figure 12.** (**Left**) Photograph of Little Bear fire on June 9 2012 (courtesy of Kari Greer, United States Forest Service. (**Right**) Photograph of the burned area.

Figure 13, depicts fire as registered by radar and imager on a satellite. The WSR-88D is at Holloman New Mexico and its code designation is KHDX. Noteworthy are the relatively high values (~ 30 dBZ) of the reflectivities. The largest Zs are at the southwest part above the location of intense burning. The fire

generated updraft lofted debris, which is highly concentrated and likely contains largest scatterers (possibly carbonated grass, or leaves etc.). The relatively low $Z_{DR}$ (about 1 dB) depicts well the updraft location and is common in plumes that are actively burning [32]. The particles on the average tend to be horizontally oriented, are likely wobbling due to turbulence and shear in the updraft, lowering the effective $Z_{DR}$. Farther downwind (to the northeast there is a secondary maximum of $Z$ coincident with a very large $Z_{DR}$, similar to observation by others [32]. We do not know the exact composition of scatterers but from the low $\rho_{hv}$, we suspect that it consists of debris in the plume. It has settled into predominantly horizontal orientation and exhibits significant wobbling. It is also possible that smoke aerosols act as condensation nuclei causing crystal formation (needles and plates) and growth [33], which further add to the reflectivity and differential reflectivity. In either case, the low values of the correlation coefficient (0.6) suggest that there is significant flutter (random canting) of the particles.

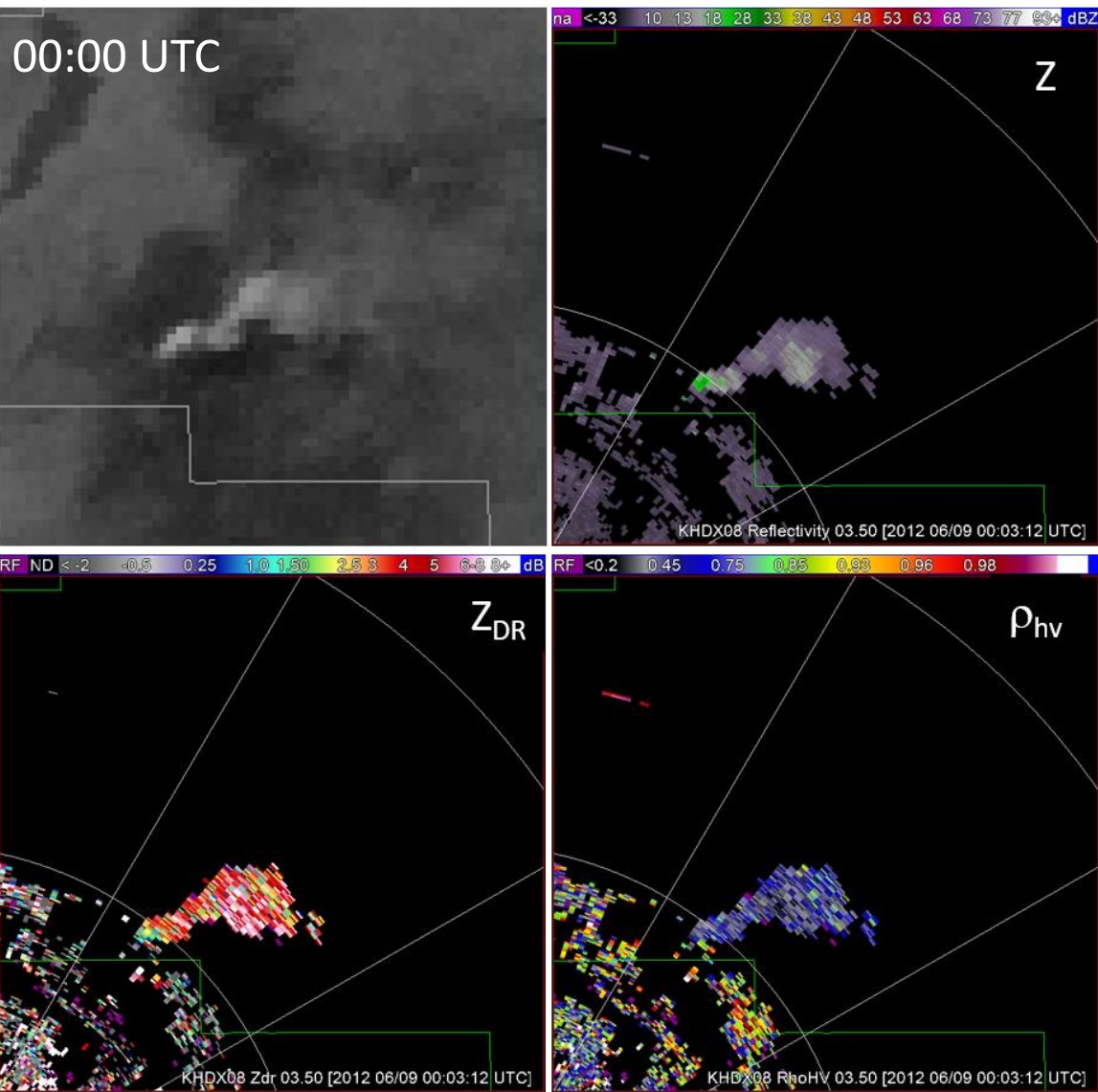

**Figure 13.** Little Bear fire on June 8 at 1800 mountain daylight timeindicated on the National Aeronautics and Space Administration (NASA) satellite photo (upper left). Field of reflectivity $Z$ from the WSR-88D (Holloman Air Force Base NM, code designation KHDX) at the same time. Field of differential reflectivity, $Z_{DR}$. Field of the correlation coefficient, $\rho_{hv}$. The elevation angle is 3.5° and time is 23:56 UTC. The range rings are at 30 and 60 km.

Refractivity variations caused by the fire could also contribute to the reflectivity [18] and influence the other polarimetric variables. From Equation 1, it follows that the structure parameter of refractivity variations $C_n^2$ should be $10^{-9}$ m$^{-2/3}$ to create a 25 dBZ return. This is two orders of magnitude larger than the value cited as "very intense" maxima observed in the boundary layer [34]. It is unknown if fires can create such large values. Fires might create much smaller values comparable to natural fluctuation, which are likely present and overwhelmed by smoke debris.

In Figure 14 are plots of the vertical profiles of the polarimetric variables. These we constructed from the conical scans and the line in Figure 14b depicts the location of the vertical slice. The returns extend over 6 km above sea level but the actual top is missing because it exceeds the height at the maximum available elevation of 4.5°. The blob of Z indicates that the particles are suspended aloft and above the boundary layer. The differential phase values are between the systems phase (about 60°) and about 100°, implying that the backscatter differential phase is between 0° and 40°. This means that some scatterers are in the Rayleigh regime while other may be oriented and inducing coupling. The differential reflectivities exhibit values in excess of 8 dB. The correlation values are between 0.6 and 0.7, indicating preponderance of nonmeteorological scatterers. The spectrum widths away from the source of smoke (i.e., fire) are smaller than 4 m s$^{-1}$; closer to the updraft the values are 8 m s$^{-1}$. Turbulence at the transition from the updraft and the environmental flow has likely caused these spectrum widths. At the location of largest spectrum width, there is a local 25 m s$^{-1}$ maximum of Doppler velocity away from the radar (not shown). This may be the beginning of the divergent flow at the top of the plume. It is interacting with the environmental wind and creating turbulence.

To determine the top of the plume we took data from the WSR-88D at Albuquerque, New Mexico, which is about 210 km away from the plume (Figure 15). The plume appears only at the 0.5° and 1.5° elevation scans and the maximum of reflectivities are 22 dBZ and 16 dBZ. Note that the $Z_{max}$ in Figure 14a is about 30 dBZ, clearly larger because at the close range of the Holloman radar smoke particles fill its beam. The beam center of the Albuquerque radar at the 1.5° elevation is 10.2 km above sea level and the lateral beam width is about 3.75 km. If the plume fills the lower part of the beam, it follows that its height would be about 8 km. The lifted condensation level on that day was 5.3 km. Therefore, the updraft likely created a cloud in which ice crystals coexist with smoke.

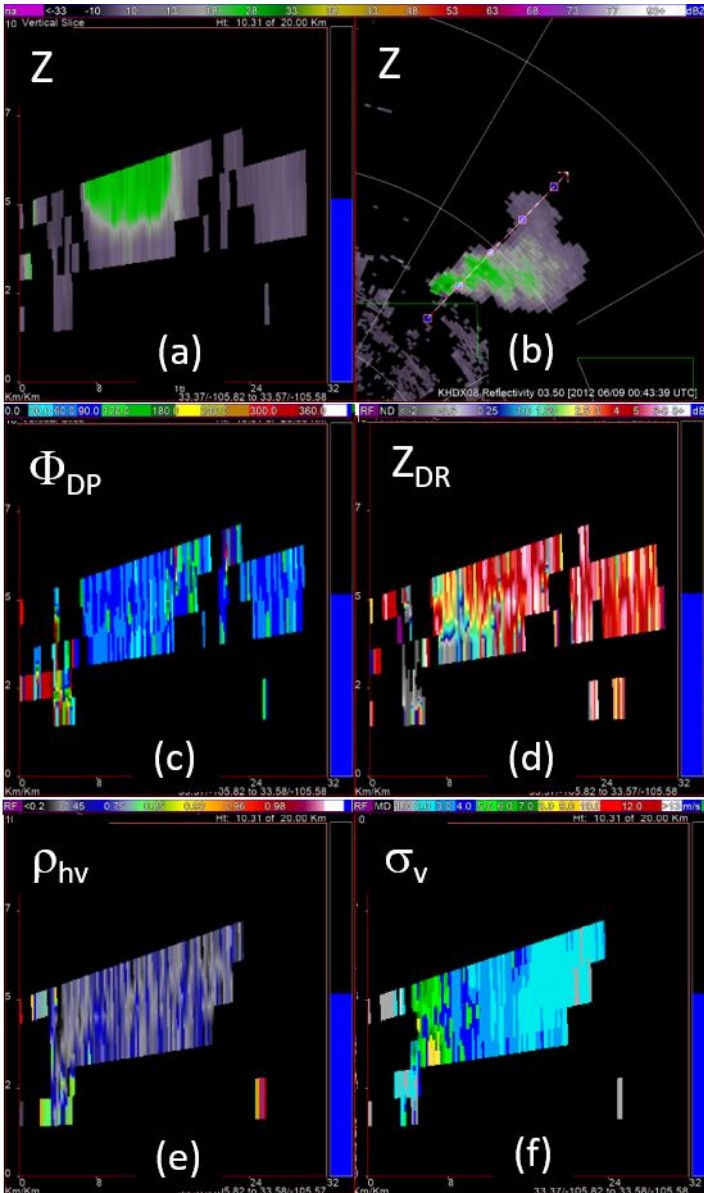

**Figure 14.** (**a**) Vertical cross section of $Z$ through the plume. (**b**) Field of $Z$ plotted as PPI (Plan Position Indicator) at the elevation of 3.5°, the arrow indicates location of the vertical cross sections. (**c**) Same as in (**a**) except the differential phase is shown. (**d**) Same as in (**a**) but the cross section is of differential reflectivity. (**e**) RHI of the correlation coefficient. (**f**) Same as in (**a**) but the plot shows the spectrum width. The top color bar indicates values of reflectivity (dBZ), the second from top color bar (above the $\Phi_{DP}$ field) indicates the values of differential phase (deg), the third color bar (above the $Z_{DR}$ field) depicts values of $Z_{DR}$ (dB), the color bar above the field of $\rho_{hv}$ indicate its categories, and the bar above $\sigma_v$ depicts values of the spectrum width. The ground altitude at the fire location is 2.9 km above mean sea level. Radar time is 00:43 UTC and the range rings are at 30, 60 and 90 km.

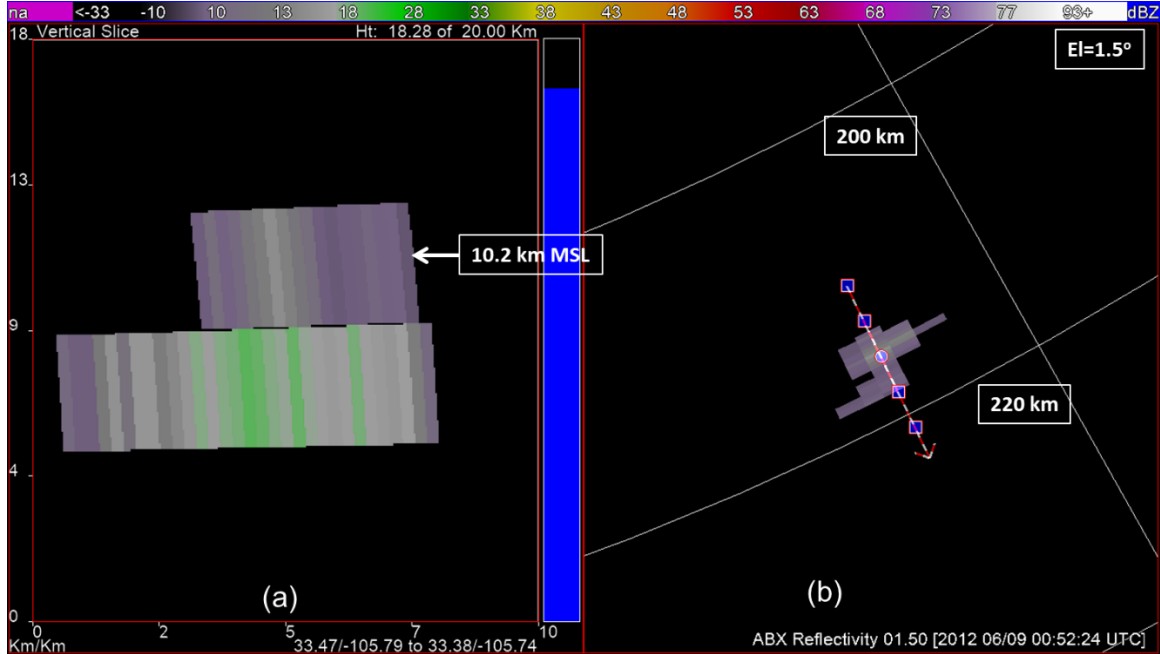

**Figure 15.** (**a**) Vertical cross section of reflectivity obtained with the Albuquerque WSR-88D (in New Mexico, code designation KABX). (**b**) Reflectivity field at the 1.5° elevation scan. The line with the arrow indicates the radial along which the vertical cross-section in (**a**) is plotted. The color bar indicates the reflectivity values (dBZ) and the time is 00:52 UTC. The range marks encompass the smoke.

Scattergrams of $Z$, $Z_{DR}$, and $Z$, $\rho_{hv}$ (Figure 16) are contained within approximately rectangular domains, suggesting that these variables are independent. Hence, for fuzzy logic type classification, the one-dimensional membership functions (at least for these variables) would suffice.

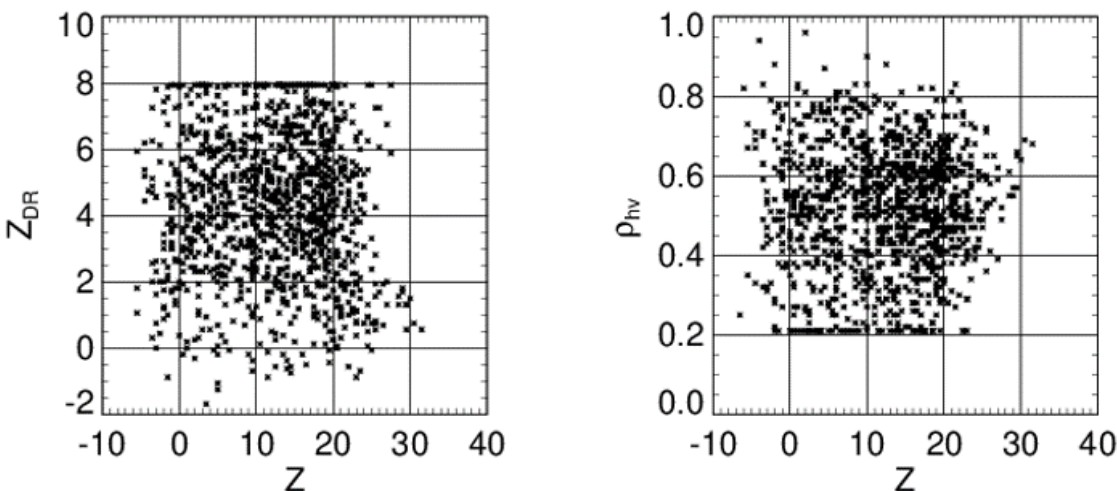

**Figure 16.** Scattergrams: (**left**) of $Z_{DR}$, $Z$, and (**right**) $\rho_{hv}$, $Z$ from the smoke plume in New Mexico.

We alert readers that the $Z_{DR}$ of 8 dB is the maximum that is currently possible to record on the WSR-88D (plans are to extend the maximum values).

Histograms of the data from the fire's patch (manually identified), indicate the mean values and spread (Figure 17). The mean values are about 12.5 dBZ reflectivity, 4.5 dB differential reflectivity, and 0.5 correlation coefficient. We have plotted also the total differential phase. Its mean value of about 60° represents the system differential phase; that is, the differential phase encountered in the transmission chain and reception chain. The spread about the mean is mostly from the variation of

the mean. The standard deviation of the estimates is approximately 2.5. The radial velocity $v_r$ of about 6 m s$^{-1}$ represents the advection component, and the spectrum width $\sigma_v$ up to 3 m s$^{-1}$ suggests presence of some turbulence.

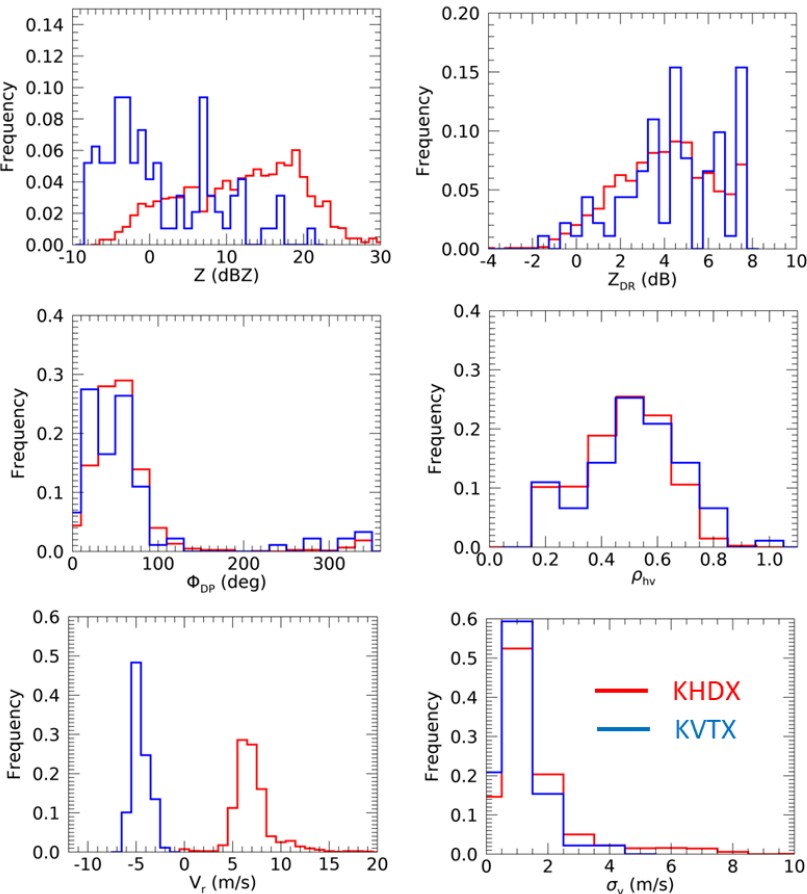

**Figure 17.** Histograms of the polarimetric variables. The red graphs stand for the New Mexico smoke plume (KHDX WSR-88D, el = 3.5°, and time is 00:43 UTC). The blue graphs are from the Los Angeles brush fire obtained with the WSR-88D, code name KVTX, el = 2.5° (time is 21:58 UTC). Date is June 17, 2017.

### 2.4. Brushfire Near Castaic Lake, California

This fire started at 13:55 PDT, on June 17, 2017. The fire burned about 800 acres of brush before it was contained a week later. The fields of polarimetric variables and Doppler velocities from smoke plume have values similar to the ones corresponding to the background consisting of biological scatterers (Figure 18). The principal distinction is that the smoke is isolated outside the range where biological scatterers are present. It is likely the biological scatterers (insects) are close to the ground and, therefore, the beam at range larger than about 40 km overshoots them.

Histograms of the polarimetric variables from the two plumes are in Figure 17. With the exception of reflectivity, the histograms of the polarimetric variables from the two events are very similar. Most Doppler spectrum widths are contained in the 0 to 2 m s$^{-1}$ interval suggesting weak turbulence. The mean differential phase equals the system phase indicating that most scatterers are Rayleigh. The spread may come from the uncertainty in estimates, which is inversely proportional to $\rho_{hv}$. Note that the modal $\rho_{hv}$ is about 0.5 characterizing nonmeteorological scatterers. Values this small increase the uncertainly of all polarimetric variables [27]. Positive $Z_{DR}$s prevail as expected from horizontally oriented small scatterers. However, there are negative values, which, we speculate, are caused by small vertically oriented smoke debris.

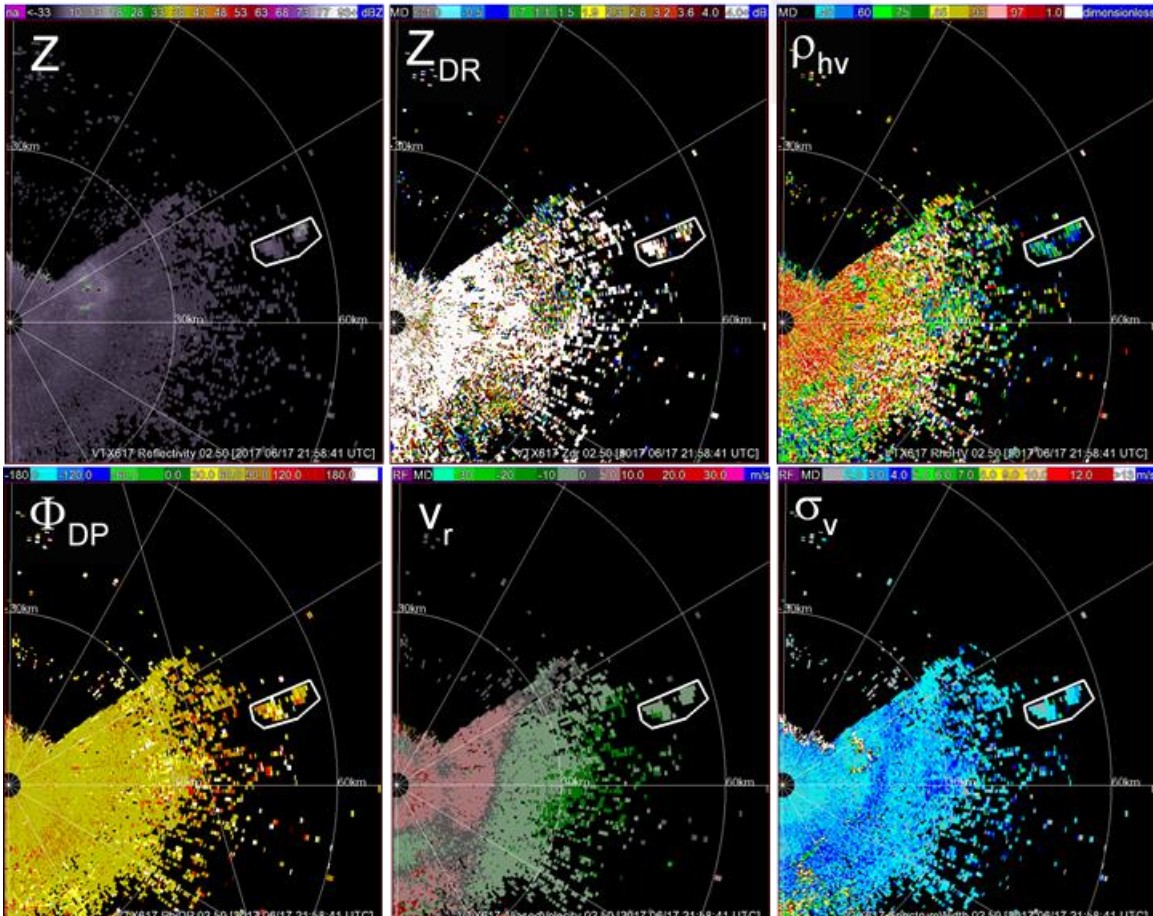

**Figure 18.** The fields of the polarimetric variables at the time of the brush fire near Castaic Lake, California, on June 17, 2017. The radar is a WSR-88D in Los Angeles (code designation KVTX). The polygon encircles the smoke plume, the elevation angle is 2.5°, the time is 21:58 UTC, and the range marks are at 30 and 60 km.

The histograms of reflectivity in the case of Little Bear fire is skewed toward larger $Z_e$s (peak is at about 20 dBZ) and maximum values reach 30 dBZ. The $Z_e$ histogram from the brush fire is skewed toward smaller $Z_e$s, with the average of about 0 dBZ. Considering that the Little Bear fire was consuming forest and was strongest, it may have lofted larger debris causing increase in reflectivity. Further, the Little Bear fire created significant updraft, which may have triggered condensation and ice crystal growth that would increase the reflectivity [8]. Doppler velocities exhibit similar spread and the mean values differ because of geometry.

## 3. Discussion

This section is about comparison of results and implication for identifying plumes among nonmeteorological scatterers. In addition to the cases presented thus far, we consider the case from [26] and two more; one is from a New Jersey forest fire and the other is from a Florida forest fire. Although we have analyzed these in detail, for brevity we only present the final results.

### 3.1. Comparisons

In Table 1, we list characteristic values of some of the polarimetric variables from eight smoke plumes. The first two in the list are the ones from Oklahoma presented in this paper. The Oklahoma case of March 12, 2008 is analyzed in [26]. The fourth case is from Kansas. In Section 2.3 we presented

the New Mexico forest fire and in 2.4 the California brush fire. The seventh and eight cases we have added to broaden the geographic span of wildfires, but did not discuss so far in this paper.

**Table 1.** Fires with the corresponding radar variables and distances.

| Date | Z (dBZ, Peak) | $Z_{DR}$ (dB) Span | Location of $\rho_{hv}$ Peak (in The Plume) | Location of $\rho_{hv}$ Peak (Outside of The Plume) |
|---|---|---|---|---|
| 12 February 2017 | 27 | −5 to 8 | 0.2 | 0.8 |
| 18 April 2017[a] | 39 | −5 to 8 | 0.2 | 0.5 |
| 12 March 2008[b] | 30 | −5 to 8 | 0.3 | 0.85 |
| 8 January 2020[c] | 30 | −5 to 8 | 0.5 | flat |
| 8 June 2012[d] | 30 | −2 to 8 | 0.5 | 0.9 |
| 17 June 2012f[e] | 20 | −2 to 8 | 0.5 | 0.9 |
| 31 March 2019[f] | 30 | −5 to 8 | 0.5 | 0.9 |
| 3 March 2019[g] | 30 | −2 to 8 | 0.5 | 0.9 |

[a]OK (Oklahoma) prairie, OK prairie analyzed in [26], [c]KS (Kansas) prairie, [d]NM (New Mexico) forest, [e]CA (California) brush, [f]NJ (New Jersey) forest, [g]FL (Florida) forest.

Note that the maximum Z can be as high as 39 dBZ. The Z histogram from the prairie fire (Figure 5) is compatible with the one from the forest fire in New Mexico (Figure 15). Both have a peak close to 20 dBZ and the maximum values of about 28 dBZ (Figures 5 and 15).

The differential reflectivities range mostly from −5 dB to over 8 dB (the truncation of recorded data). Nonetheless, in three cases the lowest value is −2 dB. The shape and spread of $Z_{DR}$ histograms in all cases of Table 1 (not shown) are very similar. The spread is considerably larger than the spread reported in [11], and so are the maximum values. The spread is also larger than found in smoke from an apartment fire [20]. The histograms overlap those caused by birds and insects.

The peaks positions of $\rho_{hv}$ from the plumes are between 0.2 and 0.5 but outside of the plumes they are between 0.5 (one case) and 0.8 to 0.9 in all but one case. Note that the peak's position of the probability density reported in [11] is very close to the positions in the two Oklahoma wildfires. However, the radar wavelength in [11] is about 3 cm; hence, many scatterers may be in the Mie regime. Consequently, their backscatter differential phases may have decreased the $\rho_{hv}$. Clearly further investigation is in order.

The $\rho_{hv}$ obtained from insects in Kansas (not shown herein) has a flat histogram, which totally overlaps the one from the plume. If the separation of the histograms peaks of insects from the ones of wildfires is sufficiently large (like in the first three cases, Table 1) separating the plume from the background returns is easier. Histograms of $\rho_{hv}$ from most wildfires are very close to the histogram from the apartment fire [20]. Note that the estimates of $\rho_{hv}$ may be biased (see the Appendix A).

We found that the mean of the backscatter differential phase for all cases is small (few degrees). The spread of the backscatter differential phase from all but the Oklahpoma February 12, 2012 case is about 50° (1 sigma width of histogram). In the OK February 12, 2012 case, the spread is about 70°. These values overlap those from the environment and this makes automatic separation challenging.

Although the average values are comparable, the spreads and shapes of the histograms do not match across the board. The polarimetric variables from the two prairie fires in Oklahoma are very similar and unique in the values of the correlation coefficient and backscatter differential phase. The $\rho_{hv}$ histograms are skewed toward zero and the histograms of $\Phi_{DP}$ are considerably wider than the corresponding histograms in the other cases. We remind readers that the backscatter differential phase matters and is equal to the difference $\Phi_{DP} - <\Phi_{DP}>$, where the brackets signify the average value, which, in the case of sparse scatterers (as here), is equal to the system differential phase. The dominant low values of $\rho_{hv}$ and the wide spread of backscatter differential phase we attribute to continuous

quick reorientation (i.e., thumbing, fluttering, spinning) of the smoke debris. This is also the reason that the correlation coefficient from smoke is generally smaller than the one from insects.

The smoke plume produced by the forest fire had a strong updraft, which lofted scatterers to about 8 km mean see level well above the liquid condensation level. Therefore, we expect that the updraft crated some cloud particles that mixed with the smoke scatterers. The prairie and the brush fires brought the smoke to the top of the planetary boundary layer, but no further.

*3.2. Classification*

We modified the simple fuzzy logic classifier [22] that separates the meteorological from the nonmeteorological returns as follows. Within the nonmeteorological category, we identify smoke plumes based on the case studies herein by assigning wildfire class to $\rho_{hv}$ between 0.2 and 0.7, and the texture (local standard deviation) of $\Phi_{DP}$ to be larger than 90°. Prior to using these thresholds, we apply a two dimensional median filter with size 17 range location by 17 radials. The $Z$, $Z_{DR}$, values from insects/birds totally overlap those of plumes. However, the local textures may have some value, which we are further researching. In Figure 19d are classified returns. Most are from biological scatterers (gray areas) and the plume (red area) is recognized fairly well. Nevertheless, there is likely a misclassification in the northeast sector.

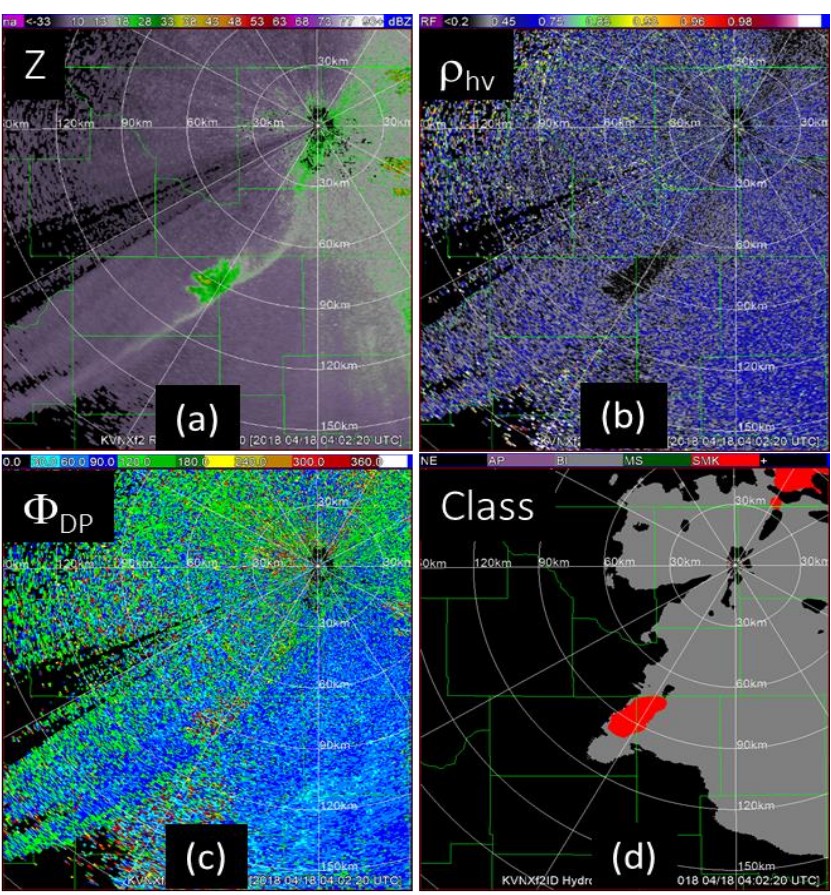

**Figure 19.** Fields of $Z$, $\rho_{hv}$, $\Phi_{DP}$, and (**d**) the results of a rudimentary classification (red color). Data is from the April 18, 2019 wildfire case in Oklahoma (time 04:02 UTC). Classes are clutter from anomalous propagation (AP), biological scatterers (BI), meteorological returns (MS), and smoke plumes (SMK). (**a**) reflectivity, (**b**) correlation coefficient, (**c**) differential phase, and (**d**) results of classification.

## 4. Conclusions

We have documented polarimetric radar observations of smoke plumes caused by wildfires of different origin. Two observations are from grass fires in Oklahoma, one is a shrub fire in California,

and one is a forest fire in New Mexico. We contrast the histograms of the polarimetric variables from these plumes to the histograms from the scatterers in the planetary boundary layer background. Moreover, we have tabulated characteristic values of $Z$, $Z_{DR}$, and $\rho_{hv}$ from the analyzed four cases and an additional four events to add to the database. The peaks of the $\rho_{hv}$ histograms from three Oklahoma grass fires are at the $\rho_{hv} \leq 0.3$. However, the histogram peak from a wild grass fire in Kansas is at $\rho_{hv} = 0.5$, which is also the location of the peaks from all other wildfires we examined. We have no satisfactory explanation for this occurrence and call for further study on a larger sample.

In the case of the forest fire, a pattern of the polarimetric variables within the plume was evident. Just in and above the pyro updraft, the $Z_{DR}$ is smaller than 1 dB and $Z$ has a maximum. In the descending region of the plume, the $Z_{DR}$ is positive and $Z$ is smaller. Others have reported similar observations. Most of our data is from the descending region in the plume.

We have compared reflectivities measured with a 10 cm wavelength WSR-88D to the ones measured with a 5 cm TDWR to infer the dominant sizes of scatterers. We use Computational Electromagnetics (CEM) tools to model scatterers in the plume and deduce sizes and orientation of the dominant ones. Our model of a fluttering ash piece as a pentagonal plate can explain the difference between the reflectivities at 10 and 5 cm wavelengths. A similar model of a straw with a hollow cylinder underestimates the magnitude of the difference. Our plate model is not sensitive to the expected thickens (0.15 to 0.25 mm). It is also insensitive to the range of permittivities expected in ash because the effect on the dielectric factor $K_m$ is modest. Therefore, the ensuing polarimetric variables are also relatively insensitive to permittivity. As we have no direct observation of the ash particles, we speculate that the plate like scatterers may be from forbs' leaves.

As an aside, we also compare background reflectivities in the planetary boundary layer of Oklahoma and conclude that the principal contributors are insects and birds. While the insects are Rayleigh scatterers at both wavelengths, the birds scatter in the Mie regime at the 5 cm wavelength.

From these observations, we constructed fuzzy logic type identifier of fire plumes within classified nonmeteorological returns. We use the correlation coefficient $\rho_{hv}$ and the texture of the differential phase. This identifier is rudimentary and requires further development with possible inclusion of additional variables.

For one Oklahoma grassfire, GOES-16 satellite data at 1 min intervals is available. A strong front observed with a WSR-88D radar blew over the fire and according to the satellite images, increased its intensity. The radar volume scans were at 10 min intervals. Because radar detection of fronts is routine, we submit that by forward extrapolating the front's position in time it may be possible to predict fire intensification. This would have significant operational implication for monitoring evolution of wildfires.

Unlike satellite, radar can observe smoke irrespective of the environmental conditions like day, night, or cloud cover. Therefore, it can serve authorities and the public for several purposes. Radar observations of smoke may provide advance information about the potential degradation of air quality. Knowing location and progression of fire can be useful to airport authorities, especially for small municipal airports. Predicting contamination of water resources by falling smoke debris, and identifying barren regions prone to mudslides is another useful information present in the data from polarimetric weather radars.

Our analysis is not comprehensive because we have examined only few aspects of wildfires. Moreover, the sample size is small. Future studies should include a comprehensive statistical analysis, detailed modeling with verifications, and further development of classification algorithm.

**Author Contributions:** Conceptualization, writing, and part of analysis, D.Z.; part of analysis and several figures, P.Z.; interpretation, theory and editing, V.M.; computational electromagnetics and part of analysis, D.M. All authors have read and agreed to the published version of the manuscript.

**Funding:** This study was provided in part by the NOAA/Office of Oceanic and Atmospheric Research under NOAA-University of Oklahoma Cooperative Agreement NA17RJ1227 US Department of Commerce.

**Acknowledgments:** Kari Greer, USGS, for the photo of the Little Bear fire and Lindsey Richardson for information concerning the values of the system differential phase on WSR-88Ds.

**Conflicts of Interest:** The authors declared no conflicts of interest.

## Appendix A

We discuss the following three effects on the correlation $\rho_{hv}$: dwell time $T_d$, polarimetric mode of operation, and the refractive index of scatterers. The dwell time is the total time during which $M$ samples spaced by the pulse repetition time $T_s$ are collected, so $T_d = MT_s$. The variance reduction and the bias of say $\rho_{hv}$ depends on the equivalent number of independent samples $M_I$ as $M_I = 4MT_s\pi^{1/2}\sigma_v/\lambda$. The biased $\rho_{hv}$ for large signal to noise ratio is Equation (A16) of [35],

$$\rho_{hv(m)} = \rho_{hv} + \frac{\left(1 - \rho_{hv}^2\right)^2}{4M_I\rho_{hv}} \tag{A1}$$

where $\rho_{hv(m)}$ is the estimate from radar data. Clearly, the sample dependent bias is always positive. Our data with mean $\rho_{hv}$ of 0.4 to 0.5 have been obtained with the longest dwell times of about 220 ms, which is about 10% larger than the dwell time used in the Oklahoma cases (196 ms). Moreover, evaluation of (A1) with these dwell times and spectrum widths of 1 to 2 m s$^{-1}$ indicates that the bias is less than 0.07, which is considerably smaller than the separation of the histogram peaks of about 0.3. See also Figure 2 of [26], where at dwell times larger than 126 ms (number of samples 128) the bias in $\rho_{hv}$ is insignificant.

In the simultaneous (SHV) mode, the polarimetric radar transmits the H and V components simultaneously; hence, the transmitted polarization depends on the phase shift between the two components, $\psi_t$. That phase shift is unknown and depends on the individual radar mainly because each radar operates at a different frequency and therefore even if the hardware is identical the $\psi_t$s will differ. In Figure 6 of [26], a 90° change of $\psi_t$ causes a drop in the $\rho_{hv}$ from about 0.5 to 0.2 in case of needle like scatterers. Although the effect may be present in some of our data, we doubt that it is strong. This is because the effect would affect equally the $\rho_{hv}$ from both the plume and the environment, which we do not see.

Whereas the previous two effects are related to the instrument, the influence of permittivity $\varepsilon$ is a physical factor. Suffice to say that increases in the real part of permittivity enhance the polarimetric variables. For example, the increase from 15 to 30 causes a drop of $\rho_{hv}$ by about 0.1 [26].

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
