# Peer review of "Of Fire and Smoke Plumes, Polarimetric Radar Characteristics"

_atmosphere, doi:10.3390/atmos11040363_

Round 1

Reviewer 1 Report

Review of Atmosphere Manuscript Revision

“Of Fire and Smoke Plumes, Polarimetric Radar Characteristics”

The authors have made substantial changes in this manuscript revision, and have at least partially addressed the vast majority of my previous concerns. I appreciate some of the figure changes and the additions of additional cases and analysis. I have mostly minor revisions to suggest on the revision. My comments below relate to the tracked-changes version of the manuscript, which doesn’t have line numbers. Thus, text locations will be less precise.

  1. Not sure why the title isn’t just “Polarimetric Radar Characteristics of Fire and Smoke Plumes.” That would read more clearly.

  1. Page 2, 2nd paragraph - What is a “good conventional system?” Please define that term more clearly. Lightning is misspelled in this paragraph.

  1. Page 2, 3rd paragraph - Referring to references [10] and [11], the authors state, “The relevant information for our study is the probability density function [PDF] of the differential reflectivity and the correlation coefficient observed in the plume and rain.” However, they don’t discuss what about those PDFs are relevant to this study. They appear to not mention [10] again, but they do discuss [11] later. Regardless, it would be helpful to the reader to hear up front what details about the PDFs in these references are relevant to the current study.

  1. Page 12 - The last line of this page appears to be a sentence fragment (“The histograms of the polarimetric variable (Fig. 11) from this plume are similar to the ones from”). It abruptly terminates mid-sentence, then there is Fig. 11 plus caption, then immediately we jump into Section 2.3. This may be an artifact of the tracked-changes document, but I wanted to raise the issue in case there is a real problem in the typesetting.

  1. Section 3.1, 1st paragraph - Should add a couple sentences each of description about the newly added 7th and 8th cases, just to orient the reader better.

  1. End of Section 3 - Why is there a misclassification in the northeast sector? The authors need to explain this better. Figure 19’s domain should also not partially cut off the display of this misclassification. It’s hard for the reader to see what is going on up there. Overall, the identification algorithm analysis remains rudimentary, and it would be helpful to see some actual performance statistics (e.g., POD, FAR, CSS, etc.).

Reviewer 2 Report

Minor comments (note, no line numbers so I can't provide):

Final sentence of abstract: Perhaps use 'Finally', or 'To conclude' to begin last sentence.

'Researchers used operational radars to infer injection heights of smoke aerosols in Southern Georgia, USA'    'to infer' is a different font

'The study found a mean height'

'Radar observations of pyroconvection combined with lightning mapping' (check for other instances of 'lightening')

Page 9: paragraph indentation changes

Table 1: Make sure columns are aligned

'In Fig. 19 is a result of classification.'  reword

4. Conclusions.  should be 1 paragraph at beginning?

Author Response

This manuscript is a resubmission of an earlier submission. The following is a list of the peer review reports and author responses from that submission.

Round 1

Reviewer 1 Report

This paper by Zrnic et al. presents dual pol weather radar data from a couple of fires (grass, industrial, and forest fire) using the WSR-88D (S-Band) network of operational radars ran by the US Federal government.

The data presented in the study are valuable, although not outstanding as compared to more recent literature published on the matter.  The main contribution from my perspective lies in the modelling of the backscattering for the second fire presented in the manuscript. The discussion on the Bragg scattering is also interesting and contributes to an improvement in knowledge on the radar dual pol radar returns of fire-generated smoke plumes.

The MAJOR flaw in this study is the total absence of review and discussion of the current literature on the subject. It reads as if the authors only know of their own work (Melnikov et al.) while a large amount of novel and substantial work has been published outside of this single author (co-author of the present paper!). A serious literature review should have been conducted!

The authors should refer in particular to the review article of McCarthy et al. (2018) in JGR-A which covers Weather radars and smoke plumes – in details. In this paper, most of the recent work is described and cited and authors should update their knowledge using this. There are also new articles published since in 2019, which the authors should read as well.

In particular the authors should read:

Laboratory studies by Baum et al. using ash

Field studies from McCarthy et al. using dual pol portable X-Band radar

Classification of pyrometeors (McCarthy et al. 2019)

Work from Lareau and Clements using single pol but discussing dual pol

Work from Palumbo et al. (PhD thesis and conference papers)

This paper has to be completely re-written in the light of this literature and re-submitted.

Reviewer 2 Report

Review of Atmosphere 698054

I recommend rejection of this manuscript, for the following reasons:

This manuscript provides only a rudimentary analysis of what appears to be a small number of radar volumes scanning 3 different fires. There are numerous examples of far more detailed analyses of polarimetric variables near fires already in the literature, many of which are by the authors of this manuscript. In my view, the limited results of this manuscript do not add significantly to the existing knowledge base, and the authors do not make a rigorous case for why this particular study is necessary.

The authors mainly cite their own work in this manuscript, which does not reflect well on their understanding of the current state of the art in polarimetric radar analysis of wildfires. Missing from the discussion and reference section are a whole host of papers by Baum, Dowdy, Lang, McCarthy, etc. The authors also do not delve deeply into other papers by some of the authors they do cite, such as Christopher, Erkelens, and Jones. Considering these many other papers as the context for the current manuscript, it becomes clear that more novel results are needed to make this manuscript publishable.

One potential source of novelty is the combined analysis of different fire types. However, the authors only consider one fire of each type, and within each category it appears they only consider 1-3 volumes. There is no statistical analysis of a broad range of different fires, and only limited consideration of plume evolution. Moreover, their case studies are not particularly high quality. For example, there are examples of much more extensive (and better scanned) wildfire plumes in the literature, compared to the Little Bear fire. There are hundreds if not thousands of wildfires in range of NEXRADs throughout the US every year. The authors have really limited themselves with their case selections. The state of the art has advanced far beyond the point where simple analysis of a few radar volumes will add anything significant to our knowledge.

Another potential source of novelty is using an extensive set of plume observations, along with scatter modeling, to develop a pyrometeor classification algorithm. The authors tease at doing this, noting in multiple locations how one could go about doing such a thing, but never follow through. Actually developing a classification algorithm would greatly improve the novelty of this manuscript.

The authors make a number of assumptions about particle types and properties, and use these assumptions in an attempt to infer the characteristics of wildfire plumes (e.g., how particles are oriented, where updrafts are located, pyrometeor size distributions, etc.). Some of these assumptions are inexplicable (e.g., assuming pentagonal leaf shapes for pyrometeors from the Oklahoma grass fire - it’s a grass fire, not a tree fire), and none are validated against in situ observations (either from the current cases or even just the literature). This leads to a lot of unjustified speculation that does not improve scientific understanding of wildfire plumes. A manuscript that drew from in situ observations to build a more complete, scattering-based model of wildfire plumes of different types, and validated that model with radar observations of multiple plumes, would be highly novel.

Figures need significant improvement, in particular the radar imagery. Very few of these figures are zoomed appropriately on the plumes themselves, making it difficult to infer the same details the authors are discussing. Moreover, the color tables are illegible. It’s unclear whether the satellite imagery is zoomed to the same scale as the radar observations, but it should be. Figures 7 and 8 add very little to the manuscript. Finally, the histograms and scattergrams do not specify which volumes are included.

Reviewer 3 Report

Major comments:

1. Line 127:  Interesting experiment…  how does shape matter?  E.g. grass blades vs. leaves.  Analysis seems incomplete with many unknowns. Have there been studies to get PSDs of smoke plumes (e.g. in-situ)? I think the end conclusion is OK, but getting there is a bit crude.

2. Lots of random white space, copy editor should make sure this gets fixed. Also make sure blurry figures are crisp for final publication.

3. Figure 15: Other figure was black/red vs. blue/red here. Any reason why all 3 fires aren’t compared? Seems strange to connect two fires on one plot, vs all three. Overall this leads to a more disjointed nature of the paper, my primary criticism.

4. Dissimilar ranges make it harder to compare fires.  For the Castaic Lake fire for example, couldn’t there be non-uniform beam filling issues? This is demonstrated in the NM fire by using two radars.

5. There should be some statements about the study being a limited sample of plumes. There have been many recent examples on WSR-88D of intense forest fire plumes, and at a minimum I know Ze would vary.

Minor Comments:

Line 15: wild fire should be one word, note this occurs in other areas of the manuscript.

Line 19: NEXRAD should be defined if used.

Line 29: geo-referenced ?

Line 40-42: Citations needed for this statement (I don’t disagree).  I would rephrase this to be less absolute.

Line 45: Perhaps mention these come from the Storm Prediction Center? I would also spend a sentence or two discussing the importance of the IMET program.

Line 52-53. I find this statement to be a stretch with the launch of GOES-16/17

Line 71: Extra ‘s’ in Los Angelas

Line 72: ‘have a wavelength of 10 cm’.

Line 73: Is there a reason why scatterers is bold?

Line 73: Mention which WSR-88D.  I would put into the format: Twin Lakes, Oklahoma  and Oklahoma City, Oklahoma

Line 78: You can reference: https://journals.ametsoc.org/doi/pdf/10.1175/BAMS-D-17-0149.1 for this dry period which was considered a flash-drought (see panel b in Fig. 1).

Figure 1: Should refer to the radar as Twin Lakes. Make sure to mention state even though it is obvious.

Line 102: Were soundings investigated to see whether they supported bragg scatter? See https://www.roc.noaa.gov/wsr88d/PublicDocs/Publications/RichardsonEtAl2017_BraggPt1.pdf

For example.

Line 105: I’d omit ‘very’

Figure  4: specify time (hours) in caption too.

Line 157/166: Yes, but citation needed.

Line 191: Is fireworks the correct word?

Line 195:/196: Sounding data could confirm this.

Line 208: Space needed in Fig.7

Line215-217: seems like an unnecessary large gap

Line 219: I would say ‘imager’ vs. camera

Line 219: Holloman, New Mexico (NM isn’t defined herein)  

Line 221: updraft ‘lofted’ debris

Line 222: updraft seems repetitive here.

Line 224.  Shear vs. sheer

Line 229: what height is the plume at? Sounding data could be useful again to understand what temperatures this is occurring.

Line 265: weird spacing here

Figure 12: Of the figures, this one is a bit blurry. As an FYI, packages like PyART could make much more visibly pleasing figures.

Line 317: Lifted? Condensation level?

Figure 14: Also rendered a bit blurry

Table 1: Caption is too short.